# Prognostic Value of Preoperative Systemic Inflammatory Parameters in Advanced Gastric Cancer

**DOI:** 10.3390/jcm11185318

**Published:** 2022-09-09

**Authors:** Sung Gon Kim, Bang Wool Eom, Hongman Yoon, Young-Woo Kim, Keun Won Ryu

**Affiliations:** 1Center for Gastric Cancer, National Cancer Center, Goyang-si 10408, Korea; 2Department of Surgery, Konyang University Hospital, Daejeon 35365, Korea

**Keywords:** systemic inflammatory parameter, advanced gastric cancer, prognosis

## Abstract

Background: The predictive value of various systemic inflammatory parameters has been reported. However, it is still unclear which inflammatory parameters are the best predictors of prognosis in advanced gastric cancer and what are their mechanisms of action. The aim of this study was to evaluate the association between preoperative systemic inflammatory parameters and overall survival (OS) in patients with advanced gastric cancer. Methods: This retrospective study included 489 patients with stage II/III advanced gastric cancer treated at the National Cancer Center, Republic of Korea, between January 2012 and December 2015. We divided the patients into survivors and non-survivors and compared their clinicopathological characteristics. Univariate and multivariate analyses using the Cox proportional hazards model were performed to evaluate the prognostic value of inflammatory parameters. Results: The absolute lymphocyte count was significantly higher in survivors (2.07 ± 0.62 × 10^3^/µL vs. 1.88 ± 0.63 × 10^3^/µL, *p* = 0.001). The neutrophil-to-lymphocyte ratio (NLR), monocyte-to-lymphocyte ratio (MLR), and platelet-to-lymphocyte ratio (PLR) were marginally lower in survivors. Survival analysis revealed that the NLR and PLR were independent prognostic factors for OS. Survival was significantly different depending on NLR and PLR in the same pathologic stages. Conclusions: NLR and PLR were independent prognostic factors for OS in patients with advanced gastric cancer. Regarding single inflammatory parameters, an elevated lymphocyte count was the only factor associated with a favorable prognosis. These results suggest that the enhanced immune function of patients affects their prognosis more than the increased systemic inflammatory response.

## 1. Introduction

Gastric cancer is a common malignant tumor and the main cause of cancer-related mortality worldwide, particularly in Eastern countries [1,2]. Radical surgical resection with regional lymphadenectomy and adjuvant chemotherapy is the mainstay of treatment for patients with advanced gastric cancer [3].

The TNM classification of the American Joint Committee on Cancer is the most common and useful prognostic tool used by clinicians that is also helpful in determining an appropriate therapeutic plan [4]. However, TNM staging alone cannot predict the exact prognosis of patients, and patients with the same stage have variable survival rates [5,6,7]. For this reason, many studies have been conducted to identify other predictive factors for various cancers to support the existing TNM staging system [8,9,10]. 

Systemic inflammation via a host–tumor interaction is recognized as the seventh marker of cancer and is closely involved in the development and metastasis of various cancers. There is increasing evidence that increased preoperative systemic inflammation is associated with disease progression and poor outcomes in advanced cancer [11,12,13]. Various parameters such as neutrophils, lymphocytes, platelets, monocytes, and albumin are known to reflect the systemic inflammatory status of patients. A previous study reported that the absolute count of blood cells, such as neutrophils, lymphocytes, monocytes, and platelets, may predict the prognosis of gastric cancer [5]. In addition, many predictive parameters have been reported using a combination of inflammatory cell counts, such as neutrophil-to-lymphocyte ratio (NLR), platelet-to-lymphocyte ratio (PLR), monocyte-to-lymphocyte ratio (MLR), and prognostic nutritional index (PNI) [14,15,16,17,18]. 

However, it is still unclear which inflammatory parameter is the best predictor of prognosis in advanced gastric cancer and what is its mechanism of action. The aim of this study was to evaluate the predictive value of preoperative systemic inflammatory parameters and their association with overall survival (OS) in patients with advanced gastric cancer.

## 2. Materials and Methods

### 2.1. Patients and Clinicopathological Data

In this single-center, retrospective study, a total of 667 patients with stage II/III advanced gastric cancer who underwent curative gastrectomy between January 2012 and December 2015 at the Center for Gastric Cancer, National Cancer Center, Republic of Korea were enrolled. We excluded patients with preoperative chemotherapy or radiotherapy (n = 13), history of other malignancies (n = 59), severe systemic diseases such as liver cirrhosis (n = 23), non-curative surgery (n = 21), immediate postoperative mortality (n = 2), and inadequate laboratory and medical records (n = 60). A total of 489 patients were included in the present study. All patients were regularly followed up until April 2020 or until death. Clinicopathological data, including age, sex, tumor characteristics, pathological stage, and operation records, were collected. We divided patients into two groups (survivor vs. non-survivor) and compared their clinicopathological data and systemic inflammatory parameters. Patients whose survival was confirmed at the time of analysis were defined as survivors, and those who died during the study period were defined as non-survivors.

### 2.2. Preoperative Systemic Inflammatory Parameters

Preoperative blood tests were performed within 14–30 days before surgery. Absolute counts of white blood cells (WBCs), neutrophils, lymphocytes, monocytes, platelets, and albumin were recorded. NLR was calculated as neutrophil count (number/L)/lymphocyte count (number/L). PLR was calculated as the platelet count (number/L)/lymphocyte count (number/L). MLR was calculated as monocyte count (number/L)/lymphocyte count (number/L), and PNI was calculated as albumin (g/dL) + 0.005 × lymphocyte count (number/L). This study was approved by the Institutional Review Board of the National Cancer Center (approval number: NCC2021-0048).

### 2.3. Statistical Analysis

Statistical analyses were performed using R version 2.12.1 (R Foundation for Statistical Computing, Vienna, Austria). To compare the differences between survivors and non-survivors, categorical variables were analyzed using the chi-square test, and continuous variables were analyzed using Student’s *t*-test and described as mean and standard deviation (SD). The ideal cutoff values of inflammatory parameters were calculated using receiver operating characteristic (ROC) curves. Univariate and multivariate analyses using the Cox proportional hazards regression model were performed to calculate hazard ratios (HRs) and 95% confidence intervals (CIs) for OS. OS was analyzed using the Kaplan–Meier method with a log-rank test. Statistical significance was set at *p* < 0.05. 

## 3. Results

### 3.1. Baseline Characteristics of Included Patients

The baseline characteristics of the patients are summarized in Table 1. The study included 334 men and 155 women. The median age was 60 years (range: 18–88 years), and the median follow-up duration was 61 months (range: 1–98 months). Of the total patients, 247 (50.5%) had stage II disease, and 242 (49.5%) had stage III disease. A total of 180 (36.8%) patients underwent total gastrectomy, and 371 (75.9%) underwent open surgery. During the study period, 182 (37.2%) patients died of cancer-related causes.

### 3.2. Difference among Preoperative Systemic Inflammatory Parameters

Preoperative inflammatory parameters were compared between the survivor and non-survivor groups and are summarized in Table 2. There were no significant differences in WBC, neutrophil, monocyte, and albumin counts among the single inflammatory parameters. Lymphocyte counts were significantly higher in the survivor group (2.07 ± 0.62 × 10^3^/µL vs. 1.88 ± 0.63 × 10^3^/µL, *p* = 0.001), and platelet counts were lower in the non-survivor group (278.1 ± 89.3 × 10^3^/µL vs. 262.4 ± 68.8 × 10^3^/µL, *p* = 0.030). There were significant differences in NLR (2.00 ± 0.85 vs. 2.30 ± 1.11, *p* = 0.003) and MLR (0.22 ± 0.09 vs. 0.25 ± 0.13, *p* = 0.002) between the survivor and non-survivor groups but a marginal difference in PLR, although this was not significant (144.4 ± 59.6 vs. 154.0 ± 60.8, *p* = 0.086).

### 3.3. Receiver Operating Characteristic Curve Analysis for Overall Survival

The ideal cutoff values of NLR, MLR, and PLR for OS were calculated based on the ROC curve (Figure 1). These parameters had statistically significant area under the curve (AUC) values (>0.5) in predicting OS, and the ideal cutoff values were set to 2.5 for NLR (*p* = 0.010), 0.2 for MLR (*p* = 0.005), and 158 for PLR (*p* = 0.004). We divided the patients into two groups based on the cutoff values and performed univariate and multivariate analyses.

### 3.4. Univariate and Multivariate Analysis of Clinicopathological Factors for Overall Survival

Univariate and multivariate analyses showed that age (≥65 years), tumor size (≥5.45 cm), depth of invasion (≥T3), lymph node invasion (≥N2), and adjuvant chemotherapy were risk factors for the prognosis of gastric cancer (Table 3). Among the inflammatory parameters, NLR and PLR were risk factors in univariate analysis. Meanwhile, NLR and PLR were independent risk factors for OS in multivariate analysis. The OS rates for stage II patients was 78.9% and that for stage III patients was 46.3%. We subsequently analyzed the prognostic value of NLR and PLR stratified by TNM stage using Kaplan–Meier analysis (Figure 2). Significant differences in OS were observed within the same TNM stage.

## 4. Discussion

Although many studies have attempted to elucidate the relationship between systemic inflammatory parameters and cancer prognosis, it is still unclear which inflammatory parameter is the best predictor for the prognosis of advanced gastric cancer and what is its mechanism of action. The results of this study were similar to those of previous studies, but while many studies focused on systemic inflammation caused by cancer progression, our results showed that the role of lymphocytes, which reflects the immune function of patients, is important. Our results suggest that the enhanced immune function of patients plays an important role in determining the prognosis of advanced gastric cancer.

The host inflammatory response to cancer cells may play a critical role in the development and progression of cancer [19]. Inflammatory mediators such as tumor necrosis factor (TNF)-α, interleukin (IL)-1, IL-6, and IL-8 are upregulated as a result of systemic inflammation [20]. These inflammatory mediators may affect changes in hematological inflammatory cells. Generally, systemic inflammation results in an increase in circulating neutrophils, monocytes, and platelets and a decrease in circulating lymphocytes. Therefore, preoperative systemic inflammatory parameters may be predictive of cancer progression and patient outcomes [21]. Many previous studies have demonstrated the predictive value of systemic inflammatory parameters in various cancers [10,20,22,23,24].

The preoperative NLR and PLR are well-known predictive parameters in various cancers, and several studies have reported the potential of these parameters to predict the prognosis of advanced gastric cancer [14,15,25,26,27]. Most previous studies have reported that elevated NLR and PLR are associated with a poor prognosis in advanced gastric cancer. Similarly, the results of the present study showed that high NLR (>2.5) and PLR (>158) are associated with poor outcomes in advanced gastric cancer. The elevated NLR may reflect an increased neutrophil-dependent inflammatory response and a decreased lymphocyte-mediated immune response against tumors, leading to tumor progression and a poor prognosis [28]. Elevated PLR is another important indicator of systemic inflammation. Platelets counts may be elevated in response to inflammatory mediators by tumors or circulating inflammatory cells, and they are considered to regulate tumor angiogenesis leading to tumor progression [20]. A previous study has reported that thrombocytosis is associated with poor prognosis in gastric cancer patients [29]. In the present study, PLR was an independent prognostic factor, and it was relevantly higher in non-survivors (144.4 ± 59.6 vs. 154.0 ± 60.8, *p* = 0.086), but unexpectedly, platelet counts were lower in non-survivors (278.1 ± 89.3 × 10^3^/µL vs. 262.4 ± 68.8 × 10^3^/µL, *p* = 0.030).

Lymphocytes play an important role in the immune response against tumors, and several subtypes of lymphocytes have the potential to eliminate cancer cells and to inhibit tumor progression by inducing antitumor response. Elevated lymphocyte counts are associated with a favorable prognosis in various cancers [5,25]. In addition, previous studies have reported that a high expression of several subtypes of tumor-infiltrating lymphocytes (TILs) such as CD8+ or CD3+ is associated with a better prognosis in gastric cancer [30,31]. Although circulating lymphocyte counts cannot accurately reflect TILs, our results show that an elevated lymphocyte count is significantly associated with a better prognosis in advanced gastric cancer, and elevated neutrophil and platelet counts were not risk factors for poor prognosis. Therefore, our results may provide additional evidence that an enhanced immune response affects the prognosis of patients with advanced gastric cancer more than the increased systemic inflammatory response.

This study has some limitations. First, it was a retrospective, single-institutional study with a limited sample size, and there might be selection bias. Further multicenter prospective studies are required to verify the results of this study. Second, the optimal cutoff values were heterogeneous among the previous studies, and the prognostic values based on different cutoff values were not compared. Third, we only analyzed the absolute number and ratio of the inflammatory parameters. However, a functional measurement of each parameter is required, especially regarding lymphocyte function.

In conclusion, NLR and PLR were independent prognostic factors for OS in advanced gastric cancer. An elevated lymphocyte count was associated with a better prognosis in advanced gastric cancer, and neutrophil and platelet counts were not risk factors for OS in the univariate analysis. This may provide evidence that enhanced immune function is a stronger determinant of prognosis than the increased systemic inflammatory response in patients with advanced gastric cancer.

## Figures and Tables

**Figure 1 jcm-11-05318-f001:**
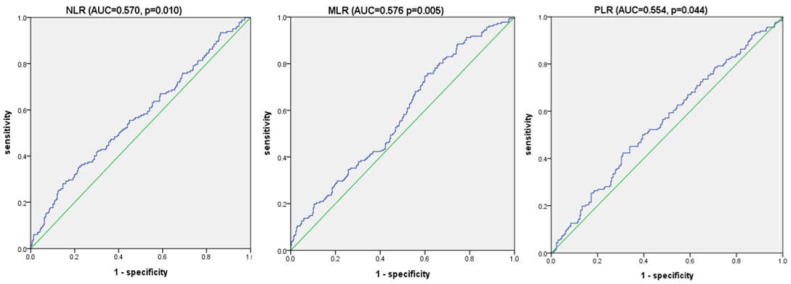
Receiver operating characteristic (ROC) curve analysis for overall survival (OS).

**Figure 2 jcm-11-05318-f002:**
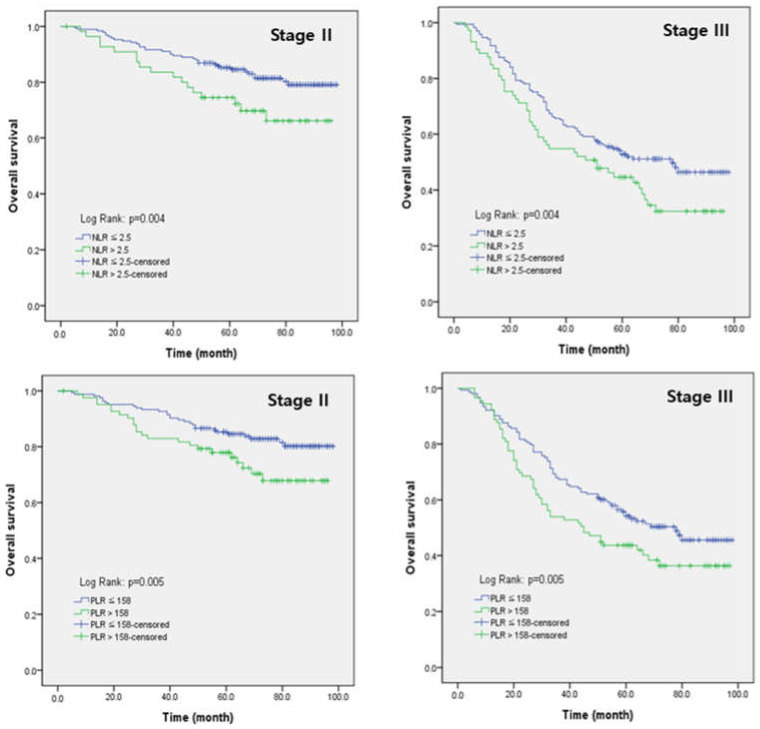
Kaplan–Meier analysis for comparison of overall survival (OS) according to NLR and PLR stratified by TNM stage.

**Table 1 jcm-11-05318-t001:** Baseline characteristics of included patients.

Variables		Total (n = 489)
Age	<65	296 (60.5%)
	≥65	193 (39.5%)
Gender	Male	334 (68.3%)
	Female	155 (31.7%)
BMI	Mean ± SD	23.5 ± 3.5
Tumor location	Upper	110 (22.5%)
	Middle	208 (42.5%)
	Lower	171 (35.0%)
Tumor size (cm)	Mean ± SD	5.93 ± 3.3
Differentiation	Differentiated	99 (20.2%)
	Undifferentiated	390 (79.8%)
T stage	1/2	110 (22.5%)
	3/4	379 (77.5%)
N stage	0/1	227 (46.4%)
	2/3	262 (53.6%)
TNM stage	II	247 (50.5%)
	III	242 (49.5%)
Operation		
Extent of resection	STG	309 (63.2%)
	TG	180 (36.8%)
Approach	Laparoscopy	118 (24.1%)
	Open	371 (75.9%)
LND	<D2	54 (11.0%)
	≥D2	435 (89.0%)
Harvested lymph nodes	Mean ± SD	50.0 ± 16.1
Complications	No	402 (82.2%)
	Yes	87 (17.8%)
Hospital stay	Mean ± SD	9.71 ± 10.0
Adjuvant chemotherapy	No	59 (12.1%)
	Yes	430 (87.9%)
Survival	Survivor	307 (62.8%)
	Non-survivor	182 (37.2%)

STG, subtotal gastrectomy; TG, total gastrectomy; LND, lymph node dissection.

**Table 2 jcm-11-05318-t002:** Difference among preoperative systemic inflammatory parameters.

Variables	Survivor (n = 307)	Non-Survivor (n = 182)	*p*-Value
WBC (10^3^/uL)	6.61 ± 1.91	6.47 ± 1.90	0.427
Neutrophil (10^3^/uL)	3.95 ± 1.54	3.97 ± 1.53	0.891
Lymphocyte (10^3^/uL)	2.07 ± 0.62	1.88 ± 0.63	0.001
Monocyte (10^3^/uL)	0.43 ± 0.16	0.44 ± 0.19	0.463
Platelet (10^3^/uL)	278.1 ± 89.3	262.4 ± 68.8	0.030
Albumin (g/dL)	4.44 ± 0.40	4.46 ± 0.40	0.771
NLR	2.00 ± 0.85	2.30 ± 1.11	0.003
MLR	0.22 ± 0.09	0.25 ± 0.13	0.002
PLR	144.4 ± 59.6	154.0 ± 60.8	0.086
PNI	24.2 ± 7.7	24.3 ± 7.6	0.879

NLR, neutrophil-to-lymphocyte ratio; MLR, monocyte-to-lymphocyte ratio; PLR, platelet-to-lymphocyte ratio; PNI, prognostic nutritional index.

**Table 3 jcm-11-05318-t003:** Univariate and multivariate analysis for overall survival (OS) using Cox proportional hazard model.

Variables		Univariate Analysis		Multivariate Analysis	
		HR (95% CI)	*p*-Value	HR (95% CI)	*p*-Value
Age(year)	≥65	1.658 (1.240–2.218)	0.001	1.715 (1.265–2.324)	0.001
Tumor size(cm)	≥5.45	1.877 (1.394–2.527)	0.000	1.433 (1.055–1.945)	0.021
Depth of invasion	≥T3	2.067 (1.355–3.153)	0.001	2.349 (1.520–3.629)	0.000
LN metastasis	≥N2	2.540 (1.844–3.498)	0.000	2.633 (1.896–3.657)	0.000
LND	<D2	1.185 (0.727–1.925)	0.499		
Adjuvant Chemotherapy	yes	0.590 (0.387–0.898)	0.014	0.602 (0.396–0.914)	0.017
NLR	>2.5	1.708 (1.258–2.318)	0.001	1.459 (1.070–1.989)	0.017
PLR	>158	1.553 (1.157–2.085)	0.003	1.504 (1.115–2.029)	0.008
MLR	>0.2	1.315 (0.970–1.783)	0.078		

HR, hazard ratio; CI, confidence interval; LN, lymph node; LND, lymph node dissection; NLR, neutrophil-to-lymphocyte ratio; PLR, platelet-to-lymphocyte ratio; MLR, monocyte-to-lymphocyte ratio.

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
