# Peer review of "Prognostic Value of Preoperative Systemic Inflammatory Parameters in Advanced Gastric Cancer"

_jcm, 2022, doi:10.3390/jcm11185318_

Round 1

Reviewer 1 Report

This in a interesting study It is a retrospective analysis of survival rates of advanced gastric cancer 489 cases were included from a single centre and followed up for at least 5 years They found that a high  lymphocyte was associated with  a good outcome.They also found that the neutrophil lymphocyte ratio and the platelet lymphocyte ratio were independent prognostic  factors 

How does your overall  survival rates for advanced gastric cancer compare to published literature   

Was stage 2 looked at by 2a or 2 b or stage 3 as 3a 3b or 3c

Would a high  lymphocyte predict a better outcome to chemotherapy or to lymph node dissection or both

Was H pylori looked for

Author Response

Thank you for your kind comments. We will answer to your comments.

1. The overall survival rate of the patients included in this study was comparable to those of previously published studies, with stage 2 (2a+2b) being 78.9 % and stage 3 (3a+3b+3c) being 46.3% confirmed. We did not separate the stages because of the complexity of the analysis. 

Would a high lymphocyte predict a better outcome to chemotherapy or to lymph node dissection or both

Was H pylori looked for

2. In this study, we didn’t analyze what you mentioned (outcome to chemotherapy or to LND and H.pylori). I think it's an interesting topic, but it needs further analysis.

Thank you for your consideration.

Reviewer 2 Report

Dear sir or madame,

thank you for the interesting manuscript. I think the scientific value of this study is present to several investigations.

I am not sure, if it is enough to write "We divided the patients in two groups (surviver vs. nonsurvivor)"

What does survivor mean? Overall? 6 months? 5 years? I could not find it in the text. I think this is absolutely necessary to discuss this topic and to report it more clearly in the manuscript.

Author Response

Thank you for your kind comments, we'll answer to your comments.

1. Survivor means overall survival. In this study, patients whose survival was confirmed at the time of analysis were defined as survivors, and those who died during the study period were defined as non-survivors. It has been added to the manuscript.

Thank you for your consideration.